# Palmitic Acid Regulation of Stem Browning in Freshly Harvested Mini-Chinese Cabbage (*Brassica pekinensis* (Lour.) Rupr.)

**DOI:** 10.3390/foods12051105

**Published:** 2023-03-05

**Authors:** Hongdou Gao, Shixian Zeng, Xiaozhen Yue, Shuzhi Yuan, Jinhua Zuo, Qing Wang

**Affiliations:** 1Key Laboratory of Vegetable Postharvest Processing, Ministry of Agriculture and Rural Affairs, Beijing Key Laboratory of Fruits and Vegetable Storage and Processing, Key Laboratory of Biology and Genetic Improvement of Horticultural Crops (North China) of Ministry of Agriculture, Key Laboratory of Urban Agriculture (North) of Ministry of Agriculture, Institute of Agri-food Processing and Nutrition, Beijing Vegetable Research Center, Beijing Academy of Agriculture and Forestry Sciences, Beijing 100097, China; 2College of Horticulture, Fujian Agriculture and Forestry University, Fuzhou 350002, China; 3College of Life Sciences, Dalian Minzu University, Dalian 116600, China

**Keywords:** mini-Chinese cabbage, palmitic acid, stem browning, antioxidant enzyme, phenolics, flavonoids

## Abstract

The effect of palmitic acid (PA) on stem browning was investigated in freshly harvested mini-Chinese cabbage (*Brassica pekinensis*). Results indicated that concentrations of PA ranging from 0.03 g L^−1^ to 0.05 g L^−1^ inhibited stem browning and decreased the rate of respiration, electrolyte leakage, and weight loss, as well as the level of malondialdehyde (MDA) in freshly harvested mini-Chinese cabbage stored at 25 °C for 5 d. The PA treatment enhanced the activity of antioxidant enzymes (ascorbate peroxidase (APX), catalase (CAT), peroxidase (POD), 4-coumarate:CoA ligase (4CL) and phenylalamine ammonia lyase (PAL)), and inhibited the activity of polyphenol oxidase (PPO). The PA treatment also increased the level of several phenolics (chlorogenic acid, gallic acid, catechin, p-coumaric acid, ferulic acid, p-hydroxybenzoic acid, and cinnamic acid) and flavonoids (quercetin, luteolin, kaempferol, and isorhamnetin). In summary, results indicate that treatment of mini-Chinese cabbage with PA represents an effective method for delaying stem browning and maintaining the physiological quality of freshly harvested mini-Chinese cabbage due to the ability of PA to enhance antioxidant enzyme activity and the level of phenolics and flavonoids during 5 d.

## 1. Introduction

Mini-Chinese cabbage (*Brassica pekinensis* (Lour.) Rupr.) is a green leafy vegetable in the family Cruciferae [1]. It is a common component of Asian diets and is becoming increasingly used in Western diets [2,3]. It has great health benefits, including anticancer, anti-obesity, and antioxidant effects [2,4]. Freshly harvested mini-Chinese cabbage, however, is very susceptible to browning, vitamin loss, softening, and the production of off-flavors, which decline its economic value [5,6]. Stem browning represents a major factor affecting the quality of freshly harvested mini-Chinese cabbage, reducing its appearance and consumer acceptance [7]. Thus, stem browning reduces the marketable shelf life of mini-Chinese cabbage. In addition to appearance, stem browning also affects the flavor of mini-Chinese cabbage, rendering it inedible.

Browning is one of the most significant defects of leafy vegetables [8]. Polyphenol oxidase (PPO) induces the synthesis of phenolics when leafy tissues are injured, which are then converted to quinones [9], resulting in a rapid browning reaction in leaf tissues. Plants also produce several antioxidant enzymes, phenolics, and flavonoids to counteract the excessive production of reactive oxygen species (ROS) in response to tissue injury. Excessive ROS accumulation induces the peroxidation and breakdown of unsaturated fatty acids in membrane lipids [10]. Several different treatments have been used to protect leafy vegetables from browning and maintain their quality during storage. Application of dimethyl dicarbonate has been reported to reduce browning in Chinese cabbage and N-phenyl-N-(2-chloro-4-pyridyl) urea was also reported to regulate browning in Chinese flowering cabbage [11,12]. A combination of 0.001 g L^−1^ 4-hexylresorcinol, 0.05 g L^−1^ potassium sorbate, and 0.025 g L^−1^ N-acetylcysteine was also reported to prevent browning of radish slices [13].

The use of palmitic acid (PA) to regulate browning has also been investigated in longan fruit [14,15]. PA (16:0) is one of the common saturated fatty acids in humans and can be obtained from ingested foods or synthesized from other carbohydrates, fatty acids, and amino acids [16]. PA is also the main component of fatty acids in cell membranes. Previous studies in longan fruit (*Dimocarpus longan*) found that the content of saturated fatty acids (including PA and stearic acid) increases when fruit starts to brown, which affects the structural integrity of cell membranes [14,15]. The structural changes in membranes induce the synthesis of PPO and phenolics, which accelerate browning [17]. Current studies indicate that PA content is enhanced during browning but that exogenous PA may inhibit browning. Thus, the role of PA in browning and the physiological response of plant tissues to PA requires additional investigation. While the effect of PA has been investigated in animal cell experiments, no reports have been published on the effect of PA on stem browning in freshly harvested mini-Chinese cabbage. This study’s object was to determine the effect of PA on freshly harvested mini-Chinese cabbage. We assessed the effect of PA on stem browning, respiration rate, electrolyte leakage, MDA content, antioxidant enzyme activity, and the level of phenolics and flavonoids in mini-Chinese cabbage during storage.

## 2. Materials and Methods

### 2.1. Plant Materials and Treatments

Mature mini-Chinese cabbages (*Brassica pekinensis* (Lour.) Rupr. cv ‘Xiaoqiao’, Beijing Shinong Seeds Co., Ltd., Beijing, China) were harvested from a farm in Xiaotangshan, Beijing, China. Mini-Chinese cabbages were approximately 0.25 m in height and harvested 60 d after planting. Harvested plants were transported to laboratory within 3 h. Mini-Chinese cabbages utilized in this study were 8–9 cm in diameter and free of any evidence of pests, disease, or mechanical damage. The freshly harvested mini-Chinese cabbages were divided into 4 groups; each group was 25 cabbages. Individual groups were immersed in either 0.03 g L^−1^, 0.04 g L^−1^, or 0.05 g L^−1^ PA dissolved in ethanol (Aladdin, AR, Shanghai, China) for 30 s with immersion in just ethanol serving as the control. The treated cabbages were placed in trays and air-dried for 10 min, after which the trays were covered with 0.03 mm polyethylene film (there is no hole on the packaging film). The cabbages were then placed in storage at 25 °C and 85–90% relative humidity in a constant temperature and humidity warehouse. Leaves and stems tissues were sampled at 0, 1, 2, 3, 4, and 5 d of storage and immediately ground to a powder in liquid nitrogen and stored at −80 °C until being further processed. Each experiment and each of the subsequent assays utilized three replicates, and the experiment was repeated three times (*n* = 9).

### 2.2. Weight Loss and Respiration Rate

Weight loss was measured as described by Duan et al. [18]. Weight loss was expressed as the percentage loss from the original weight and calculated using the formula:Weight loss (%) = 100% × (Initial weight − final weight)/Initial weight(1)

The respiration rate was measured using a F-940 Gas Analyzer (Felix, Washington, DC, USA). A 500 g sample of cabbage was placed in a 1 L gas-tight box for 1 h after which the concentration of CO_2_ was determined. Results are expressed as mg CO_2_ kg^−1^ h^−1^.

### 2.3. Color and Browning Index (BI)

An CR-400 automated colorimeter (Konica Minolta Holdings, Inc., Tokyo, JAPAN) was used to measure the color and browning index (BI) [19]. *L**, *a**, and *b** of sample were determined and the BI was calculated using the formula as follows:(2)$$BI=\frac{100\times (x-0.31)}{0.172}+180$$
(3)$$x=\frac{a^{*}+1.75L^{*}}{5.645L^{*}+a^{*}-3.012b^{*}}$$

### 2.4. Electrolyte Leakage and MDA Content

Electrolyte leakage was measured with a DDS-11A conductivity meter (Shanghai Instrument and Electronic Scientific Instruments, Ltd., Shanghai, China) by the method of Li et al. [12]. A leaf disc with a diameter of 1 cm was collected from ten different leaves and each disc was placed in 25 mL of distilled water (dH_2_O). Conductivity of the solution was measured after 2 h of incubation at room temperature and used as the initial value (P1). The solutions containing the leaves were then boiled at 100 °C for 5 min, cooled, and conductivity (P2) was again assessed. Percentage electrolyte leakage was calculated using the method as follows: P1/P2 × 100.

MDA content was determined by the method of Xu et al. with minor modifications [20]. Samples (1 g) were homogenized in 5 mL of 10% (*w/v*) trichloroacetic acid (TCA) (Shanghai Macklin Biochemical Co., Ltd., AR, Shanghai, China) and centrifuged at 12,000× *g* for 15 min at 4 °C. Then, 1 mL supernatant was mixed with 3 mL of 0.5% (*w/v*) thiobarbituric acid (Shanghai Macklin Biochemical Co., Ltd., Analytical reagent(AR), Shanghai, China), boiled 20 min. After this, the absorbance (UV-1800 spectrophotometer, Shimadzu, Tokyo, Japan) of the solution was determined at 450, 532 and 600 nm. MDA content was calculated (μmol L^−1^) = {[6.45 × (OD_532_ − OD_600_) − 0.56 × OD_450_] × V}/(V_S_ × m × 1000), where V is the total volume of the extracting solution, mL; Vs is the measured volume of the extracting solution, mL; and m is the weight of the sample, g.

### 2.5. Total Phenolics and Total Flavonoids

Frozen samples of cabbage (2 g) were powdered and mixed with 10 mL 80% ethanol (AR, Aladdin, Shanghai, China) (diluted by dH_2_O), and extracted for 40 min at 40 °C. The mixture was then centrifuged (D-37520 centrifuge, Beijing Chengmao Industrial Science & Development Co., Ltd., Beijing, China) at 12,000× *g* for 25 min at 4 °C. The supernatant was used for measuring total phenolics and flavonoids. Total phenolics were determined using Folin–Ciocalteu (FC) (Shanghai Yuanye Bio-Technology Co., Ltd., Shanghai, China) reagent by the method of Fan et al. [21]. Test tubes containing 800 μL dH_2_O and 200 μL FC reagent were prepared and then 400 μL supernatant was added to the test tube, vortexed, and incubated about 3 min at 20 °C. Subsequently, 400 μL 20% (*w/v*) Na_2_CO_3_ (Aladdin, AR, Shanghai, China) and 1.2 mL dH_2_O were added to the mixture. Then a water bath (XMTD-6000 water bath kettle, Yuyao Jindian Instrument Co., Ltd., Yuyao, China) was prepared at 20 °C for 60 min. Absorbance of the sample solutions at 760 nm were then measured in a spectrophotometer (UV-1800 spectrophotometer, Shimadzu, Tokyo, Japan) and used to determine the level of total phenolics based on the use of a standard curve constructed using different concentrations of gallic acid (Shanghai Yuanye Bio-Technology Co., Ltd., Shanghai, China).

Flavonoid content was measured using the method by Zhou et al. [22] with some modification. Amounts of 1 mL supernatant, 0.25 mL of 10% (*w/v*) AlCl_3_ (Aladdin, AR, Shanghai, China) and 1 mL 5% (*w/v*) NaNO_2_ (Aladdin, AR, Shanghai, China) were mixed. After 5 min, 1 mL of 1.0 mol L^−1^ NaOH (Aladdin, AR, Shanghai, China) was added to the mixture. Catechinic acid (Shanghai Yuanye Bio-Technology Co., Ltd., Shanghai, China) was used as a standard to calculate the flavonoid content. Absorbance of the resulting sample solution was measured at 510 nm and used to determine flavonoid content. The concentration of total phenolics and flavonoid content were expressed as g kg^−1^.

The level of specific phenolics (gallic acid, catechin, chlorogenic acid, p-Hydroxybenzoic acid, p-Coumaric acid, ferulic acid, and cinnamic acid) (Shanghai Yuanye Bio-Technology Co., Ltd., Shanghai, China) and specific flavonoids (quercetin, kaempferol, luteolin, and isorhamnetin) (Shanghai Yuanye Bio-Technology Co., Ltd., Shanghai, China) were determined by HPLC (1260, Agilent Technologies Co., Ltd., Palo Alto, CA, USA) according to the method by Xu et al. [20]. Briefly, 2 g sample powder were mixed with 2 mL of methanol (Aladdin, GR, Shanghai, China), which was ultrasonicated for 1 h (40 kHz), after which the mixture was centrifuged at 10,000× *g* for 15 min. The supernatants were utilized in the HPLC analysis. Conditions were as follows: utilization of an Eclipse Plus C18 (250 mm × 4.6 mm, 5 μm) column, with a temperature of 30 °C. The detector was a UV-detector (280 nm). The mobile phase consisted of 1% of formic acid-water (A) and methanol (100%) (B) and the gradient elution conditions were 0–3 min, 15% to 30% B; 3–35 min, 30% to 45% B; 35–45 min, 45% to 65% B; 45–50 min, 65–15% B.

### 2.6. CAT, APX, PPO, and POD Enzyme Activity

CAT and APX enzyme activity was assessed according to the method by Wang et al. [23]. Briefly, 2.0 g of cabbage powder was mixed with 10 mL of 0.1 mol L^−1^ phosphate buffer solution (PBS, pH 7.8 containing 0.05 g of polyvinylpyrrolidone (PVPP, Golden Clone (Beijing) Biotechnology Co., Ltd., Beijing, China)), and then centrifuged at 12,000× *g* for 10 min at 4 °C. The supernatant was utilized to determine CAT and APX activity.

Amounts of 0.1 mL of supernatant, 1 mL of 0.3% (*v/v*) dH_2_O_2_, and 1.9 mL 0.05 mol L^−1^ PBS (pH 7.8) were mixed. Absorbance of the resulting solution was determined at 240 nm to calculate CAT activity. APX activity reaction mixture included 1.2 mL supernatant, 2.6 mL of 0.1 mmol L^−1^ EDTA (Shanghai Macklin Biochemical Co., Ltd., AR, Shanghai, China) and 0.5 mmol L^−1^ AsA (Aladdin, GR, Shanghai, China) (in PBS, pH 7.5), and 0.3 mL of 2 mmol L^−1^ dH_2_O_2_. Absorbance was measured at 290 nm to calculate APX activity.

PPO and POD activity were determined by the method of Zhou et al. with slight modification [22]. Frozen cabbage powder (2 g) was added to 10 mL of 0.1 mol L^−1^ of PBS (pH 6.4), containing 0.05 g PVPP. The resulting mixture was then centrifugated at 12,000× *g* for 30 min at 4 °C. Next, 0.1 mL of supernatant was mixed with 0.6 mL of 50 mmol L^−1^ catechol (Nantong Runfeng Petrochemical Co., Ltd., AR, Nantong, China) substrate to measure PPO activity, while 0.9 mL of 0.2% guaiacol (Nantong Runfeng Petrochemical Co., Ltd., AR, Nantong, China) (*v/v*) was mixed with 1 mL of 0.3% dH_2_O_2_ (*v/v*) to measure POD activity. Absorbance was measured at 410 nm and used to calculate PPO activity and at 470 nm to calculate POD activity.

### 2.7. PAL and 4CL Enzyme Activity

The assessment of PAL activity was conducted by Kamdee et al. with slight modification [24]. Briefly, 1 g of powdered cabbage leaf tissue was added to 4 mL of 50 mmol L^−1^ borate buffer (BBS, pH 8.5) including 5.0 mmol L^−1^ 2-mercaptoethanol (Aladdin, GR, Shanghai, China) and 0.4 g PVPP. The resulting mixture was then centrifuged at 12,000× *g* for 20 min at 4 °C. An amount of 0.3 mL supernatant was added to 0.7 mL of 100 mmol L^−1^ l-phenylalanine (AR, Xiya Reagent, Linyi, China) and 3 mL of 50 mmol L^−1^ borate buffer (BBS, pH 8.5). The mixture was incubated at 40 °C for 1 h and then 0.1 mL of 5 mmol L^−1^ HCl (Beijing Institute of Chemical Reagents, AR, Beijing, China) was added to stop the reaction. Absorbance of the solution was measured at 290 nm to calculate PAL activity.

The level of 4CL enzyme activity was determined using a commercial assay kit (Comin Biotechnology Co., Ltd., Suzhou, China). Briefly, 2.0 g of sample were homogenized in 10 mL of extraction buffer, and the resulting mixture was then centrifuged at 8000× *g* for 10 min at 4 °C. Absorbance at 333 nm was measured to calculate 4CL enzyme activity.

### 2.8. Data Analysis

SPSS ver. 22 (SPSS Inc., Chicago, IL, USA) software was used to conduct the statistical analyses. Data were subjected to a two-way ANOVA and mean separations were performed using a Pearson’s multiple range test. In the case of single mean comparisons, data were subjected to an LSD analysis in which differences at *p* < 0.05 were considered significant. All results presented are means ± standard deviation (SD).

## 3. Results

### 3.1. Stem Browning

The visual color of cabbage stems is an excellent indicator of their degree of browning. The level of stem browning in the control group greatly reduced their visual quality over the 4 d of storage (Figure 1A). Cabbage stems treated with PA, however, exhibited a slower rate of browning than the control over the 4 d of storage. This was also reflected in the BI (Figure 1B). The BI increased during storage in both PA-treated and control cabbage stems; however, the level of BI was lower in PA-treated samples than it was in the control group (Figure 1B). Notably, cabbage stems treated with 0.05 g L^−1^ PA had the lowest BI relative to the control and to cabbages treated with lower concentrations of PA.

The BI is calculated from the values obtained for *L**, *a**, and *b**. Figure 1C–E illustrate the changes in *L*, a*,* and *b** in stems of PA-treated and control groups over 5 d of storage at 25 °C. The level of *L** in both control and PA-treated samples decreased during storage; however, in the control group the *L** value was lower than it was in PA-treated cabbages during the entire course of storage. In contrast, *a** and *b** exhibited an increasing trend, with the level of increase in the different treatment groups ranging from the control > 0.03 g L^−1^ PA > 0.04 g L^−1^ PA > 0.05 g L^−1^ PA.

### 3.2. Respiration Rate, Weight Loss, Electrolyte Leakage, and MDA Content

The respiration rate exhibited a rapid increase after cabbages were placed in storage, reaching its highest level after 2 d, after which the respiration rate decreased and maintained a relatively stable value (Figure 2A). Peak respiration rates were 17.752 (control), 14.063 (0.03 g L^−1^ PA), 11.782 (0.04 g L^−1^ PA), and 10.625 (0.05 g L^−1^ PA) mg CO_2_ kg^−1^ h^−1^.

The percentage weight loss increased in all groups (Figure 2B); however, the weight loss in control samples was higher than it was in any of the PA-treated samples from 1 to 5 d of storage. The percentage weight loss in the control group was 0.735%, which was higher than the 0.517% weight loss observed in the 0.05 g L^−1^ PA-treated sample group after 5 d.

Electrolyte leakage also showed an increasing trend in all groups (Figure 2C). The increase in the control group was higher than that in the PA-treated groups starting from 2 d. The 0.03 and 0.04 g L^−1^ PA-treated cabbages exhibited a similar pattern to each other and were higher than they were in the 0.05 g L^−1^ PA-treated samples. Electrolyte leakage had increased to 0.216%, 0.192%, and 0.181% in the 0.03, 0.04, and 0.05 g L^−1^ PA treatment groups, respectively, and 0.290% in the control group, after 5 d.

MDA levels increased before the first 3 d of storage in all treatment groups, and then declined (Figure 2D). Lower MDA levels were observed in PA-treated samples than in the control group. Peaks of MDA content in the 0.03, 0.04, and 0.05 g L^−1^ PA-treated samples were 1.868, 2.111, and 1.213 μmol kg^−1^, compared to the control group (3.080 μmol kg^−1^) of mini-Chinese cabbages.

### 3.3. PPO, 4CL, PAL, APX, CAT, and POD Activity

PPO activity in the control group was higher than it was in the PA-treated samples (Figure 3A), exhibiting an increasing trend during the first 3 d in all groups and then declining. Peak PPO activity was 0.363 units in the control group, 0.325 units in the 0.03 g L^−1^ PA-treated group, 0.108 units in the 0.04 g L^−1^ PA-treated group, and 0.095 units in the 0.05 g L^−1^ PA-treated group, all of which were lower than that in the control group.

The enzymatic activity of 4CL rapidly increased in the first 2 d and then fluctuated from 2 to 5 d. Higher activity was observed in the PA-treated group. After 2 d of storage, 4CL activity was 800.312, 837.459, and 828.569 units in the 0.03, 0.04, and 0.05 g L^−1^ PA-treated samples, respectively.

Changes of PAL activity in the control and PA-treated groups are shown in Figure 3C. PAL activity indicated an increasing trend in all groups during the first 2 d, with PA-treated groups generally exhibiting higher activity. The 0.05 g L^−1^ PA-treated samples exhibited the highest level of activity (93.600 units).

An increase in APX activity was exhibited in the control and PA-treated groups during the first 2 d and then declined (Figure 3D). APX activity in the control group was lower (0.036 units) at 2 d of storage, while APX activity was 0.042 units, 0.045 units, and 0.053 units in the 0.03, 0.04, and 0.05 g L^−1^ PA treatment groups, respectively.

The data indicated that CAT activity in all groups reached a maximum at 3 d and then decreased (Figure 3E). Although the pattern of CAT activity was similar in all of the treatment groups, CAT activity was higher in general in the PA-treated sample groups than it was in the control group.

POD activity reached a maximum after 2 d of storage and then remained stable (Figure 3F). POD activity after 5 d of storage was 0.185, 0.185, 0.248, and 0.288 units in the control group and the 0.03, 0.04, 0.05 g L^−1^ PA treatment groups, respectively.

### 3.4. Total Phenolics, Gallic Acid, Catechin, Chlorogenic Acid, p-Hydroxybenzoic Acid, p-Coumaric Acid, Ferulic Acid, and Cinnamic Acid

The level of total phenolics increased in both the control and PA-treated groups over the 5 d (Figure 4A). Although the trend was similar in all of the treatment groups, the levels of total phenolics in PA-treated groups were higher than they were in the control group. The level of total phenolics after 5 d of storage was 1.456 g kg^−1^, 1.863 g kg^−1^, 2.331 g kg^−1^, and 1.658 g kg^−1^ in the 0.03 g L^−1^ PA, 0.04 g L^−1^ PA, 0.05 g L^−1^ PA and control treatment groups, respectively.

Cinnamic acid content gradually increased in all groups (Figure 4B). The highest level of cinnamic acid (2.361 mg kg^−1^) was observed at 3 d in the 0.05 g L^−1^ PA-treated group.

The content of gallic acid exhibited an initial decrease at 1 d of storage and then an increasing trend in all groups from 2 to 5 d (Figure 4C). The content of gallic acid after 5 d of storage was 4.030 mg kg^−1^, 5.067 mg kg^−1^, 5.615 mg kg^−1^, and 6.873 mg kg^−1^ in the control, 0.03 g L^−1^ PA, 0.04 g L^−1^ PA, and 0.05 g L^−1^ PA treatment groups, respectively. The chromatogram and standard curve are in Appendix A.

Catechin content exhibited an increasing trend in the control and PA-treated groups during the first 4 d and then declined (Figure 4D). At 4 d, the 0.04 and 0.05 g L^−1^ PA-treated samples exhibited a higher level (20.632 and 20.293 mg kg^−1^) of catechin.

Chlorogenic acid content exhibited an increasing trend during storage, with PA-treated groups having a higher content than that in the control group (Figure 4E). Chlorogenic acid content at 5 d was 21.757 mg kg^−1^ (0.03 g L^−1^ PA), 20.866 mg kg^−1^ (0.04 g L^−1^ PA), 29.859 mg kg^−1^ (0.05 g L^−1^ PA), and 17.276 mg kg^−1^ (control).

The level of p-Hydroxybenzoic acid showed an enhanced trend during storage in PA-treated groups (Figure 4F). The level of activity in the 0.05 g L^−1^ PA treatment group exhibited a change at 3 d (7.640 mg kg^−1^), a level that was much higher than in other groups.

The level of p-Coumaric acid exhibited a slight to high increase in all groups during the first 3 d, and then declined. The level of p-Coumaric acid exhibited a peak of activity in the 0.03 g L^−1^, 0.04 g L^−1^, and 0.05 g L^−1^ PA treatment groups at 3 d of storage when levels increased to 3.818 mg kg^−1^, 3.822 mg kg^−1^, and 9.083 mg kg^−1^, while the level in the control group remained stable (Figure 4G).

The level of ferulic acid fluctuated in all groups (Figure 4H). The level of ferulic acid in the 0.05 g L^−1^ PA-treated group at 3 d was 6.973 mg kg^−1^, which was more than 1.28 times greater than it was in the other treatment groups.

### 3.5. Total Flavonoids, Quercetin, Luteolin, Kaempferol, and Isorhamnetin

Flavonoid content exhibited the same trend as total phenolics, increasing with storage time in all the treatment groups (Figure 5A). The highest content of flavonoids (1.455 g kg^−1^) was observed in the 0.05 g L^−1^ PA-treated group and was higher (1.63 times greater) than in the control group.

Quercetin content increased before the first 4 d and then decreased (Figure 5B). Quercetin content, however, was higher in the 0.05 g L^−1^ PA treatment group than it was in the control group during all the storage time.

Luteolin content increased during the first day and then decreased in all groups (Figure 5C). Peak content was observed at 2 d in 0.03 g L^−1^, 0.04 g L^−1^, 0.05 g L^−1^ PA treatment groups, at which luteolin content was 6.788 mg kg^−1^, 7.418 mg kg^−1^, and 9.109 mg kg^−1^, respectively.

Kaempferol levels greatly increased in PA-treated groups (Figure 5D). In contrast, kaempferol content changed only slightly in the control group. Kaempferol content in all of the PA-treated groups was higher than it was in the control group.

Isorhamnetin content exhibited a decreasing trend in all groups (Figure 5E). Isorhamnetin content was higher in the 0.04 g L^−1^ and 0.05 g L^−1^ PA-treated groups than it was in other groups from 1 to 3 d.

### 3.6. Correlation Analysis

Pearson coefficients were used to determine the correlation between the different measured parameters (Figure 6A). The analysis indicated that BI levels were significantly positively correlated with *a**, *b**, weight loss, electrolyte leakage, MDA content, and 4CL activity. BI levels were negatively correlated with *L** (Figure 6A). Total phenol content was significantly positively correlated with the level of cinnamic acid, gallic acid, catechin, chlorogenic acid, p-Hydroxybenzoic acid, flavonoids, quercetin and kaempferol. Total phenol content was significantly negatively correlated with the content of luteolin and isorhamnetin (Figure 6B).

## 4. Discussion

Previous studies have indicated that tissue browning in plants induces an increase in the level of PA, which may explain why exogenous PA can inhibit browning [14,15]. Mini-Chinese cabbage stems are prone to browning after cabbages are harvested and browning is the basis for a loss in quality. This study indicated that mini-Chinese cabbages treated with PA for 30 s exhibit a reduction in weight loss and the rate of respiration, and reduced levels of stem browning and electrolyte leakage. In contrast, antioxidant enzyme activity and many phenolics and flavonoid compounds were enhanced. These responses collectively helped to maintain the quality of cabbage heads in storage and extend their shelf life.

The commercial value of mini-Chinese cabbages is dependent on their quality. Thus, it is essential to maintain their quality during storage. Weight loss and high respiration rates are known indicators of quality degradation in mini-Chinese cabbage and reports have indicated that strategies that suppress respiration and reduce evaporation in harvested plants help to maintain their postharvest quality [25]. In our study, weight loss and respiration rate were reduced in response to immersion of the cabbage heads in PA for 30 s. These results confirm that PA lowers the basal metabolism of mini-Chinese cabbage. The permeability of cell membranes increases when plants begin to experience senescence, which promotes the peroxidation of lipids [10]. MDA is commonly recognized as an indicator of oxidative stress and peroxidation [26], and increased electrolyte leakage and MDA levels serve as indicators of membrane degradation and peroxidation of membrane lipids, respectively [27]. Results exhibited that PA treatment reduced both the level of electrolyte leakage and MDA levels in mini-Chinese cabbages, relative to the untreated control. These results are similar to those in a previous study by Li et al. [12], who reported that electrolyte leakage and MDA levels in Chinese flowering cabbage were maintained during storage by treatment with N-phenyl-N-(2-chloro-4-pyridyl) urea (CPPU). Additionally, PA is the main fatty acid component of cell membranes and contributes to the resilience of cell membranes exposed to stress. Our collective results indicate that PA treatment of mini-Chinese cabbage can help to maintain their quality in storage.

Browning is a complex process that can be affected by many enzymes and chemical compounds, in which tissue color is the most recognizable visual manifestation. In our study, the PA treatment delayed the process of stem browning, as indicated by a reduction in the BI in stems of mini-Chinese cabbage, relative to the control. High levels of PPO activity are known to induce enzymatic browning of plant tissues during storage by catalyzing the oxidation of phenolics to quinones [28]. In our study, PPO activity was positively correlated with BI in mini-Chinese cabbage. PAL and 4CL are enzymes that play an essential role in the synthesis of phenolics and their activity is a component of phenylpropane metabolism. Thus, increased PAL and 4CL activity can be responsible for an increase in the level of phenolics [29,30]. Our results revealed that the PA treatment enhanced both 4CL and PAL activity. This may be attributed to the ability of PA to reduce PPO activity which would have reduced the production of quinones [9]. PAL and 4CL activity would also increase the synthesis and accumulation of phenolics, resulting in enhanced antioxidant capacity [30]. Thus, the increase in PAL and 4CL activity by PA would increase antioxidant capacity. The increased accumulation of phenolic compounds would also enhance defense capacity [31]. Considerable evidence indicates that the oxidative stress resulting from excess production of ROS may induce the browning of plant tissues and that antioxidant enzymes, such as APX, CAT, and POD, play an essential role in reducing ROS levels and inhibiting browning [20,32,33,34]. In the present study, PA increased APX, CAT, and POD activity in mini-Chinese cabbage. These results confirm that PA plays a role in reducing excessive levels of ROS through its ability to enhance antioxidant enzyme capacity.

Browning is a great stress in wounding fruit and vegetables during postharvest, a significant postharvest physiological disorder. Plant cells subjected to stress respond by activating two aspects of phenolic metabolism [35]. One involves the antioxidant properties of phenolic compounds, along with antioxidant enzymes, that work together to reduce oxidative stress. The other involves the use of monomeric and polymeric phenolic compounds, whose synthesis is catalyzed by PAL in the phenylpropanoid pathway, to seal off injured tissues. These monomeric and polymeric phenolic compounds have strong antioxidant properties and are also used to form a physical barrier against invading pathogens [31]. In our study, the level of phenolic compounds (including gallic acid, chlorogenic acid, p-Coumaric acid, catechin, p-Hydroxybenzoic acid, ferulic acid, and cinnamic acid) and flavonoids (such as luteolin, quercetin, kaempferol, and isorhamnetin) were enhanced during storage by the PA treatment, relative to the control group. These compounds represent monomeric and polymeric phenolics that play a role in inhibiting stem browning through their ability to scavenge ROS [36] and inhibit lipid oxidation [37]. Both activities would inhibit stem browning in min-Chinese cabbages. PA enhanced the level of both total phenolics and flavonoids, and the enhanced level of these compounds was associated with reduced stem browning in mini-Chinese cabbages during storage at 25 °C.

## 5. Conclusions

Treatment of mini-Chinese cabbages with PA greatly inhibited weight loss, reduced the rate of respiration, and inhibited stem browning, which collectively served to maintain the overall postharvest quality of mini-Chinese cabbage stored at 25 °C for 5 d. The mechanism underlying the inhibition of stem browning in mini-Chinese cabbage by PA was associated with decreased levels of membrane damage, as evidenced by lower levels of electrolyte leakage and MDA, and an increase in antioxidant metabolism, as evidenced by higher antioxidant enzyme activity, and elevated levels of phenolics and flavonoids. The collective results of the present study determine that the application of PA has the potential to be used as a method to maintain the quality and to extend the shelf life of mini-Chinese cabbage after harvest and during storage.

## Figures and Tables

**Figure 1 foods-12-01105-f001:**
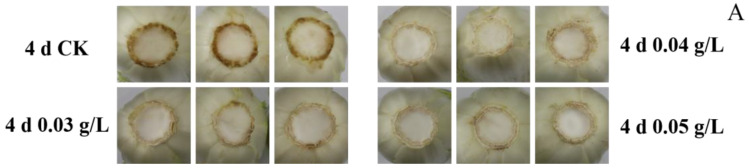
Effect of different concentrations of PA on the appearance (**A**), browning index (**B**), *L^*^* (**C**), *a^*^* (**D**), *b^*^* (**E**) of mini-Chinese cabbage during five days of storage at 25 °C. Data represent the mean ± SD (*n* = 9). Different letters above the bars within each time point represent a significant difference between the designated treatment groups at *p* < 0.05 as determined by an LSD test.

**Figure 2 foods-12-01105-f002:**
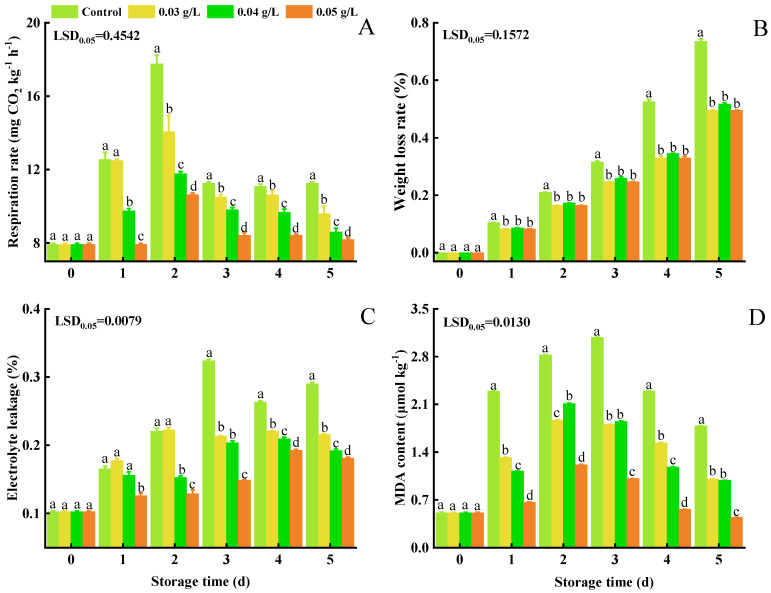
Effect of PA on respiration rate (**A**), weight loss (**B**), electrolyte leakage (**C**), and MDA content (**D**) in mini-Chinese cabbage during five days of storage at 25 °C. Data represent the mean ± SD (*n* = 9). Different letters above the bars within each time point indicate a significant difference between the designated treatment groups at *p* < 0.05 as determined by an LSD test.

**Figure 3 foods-12-01105-f003:**
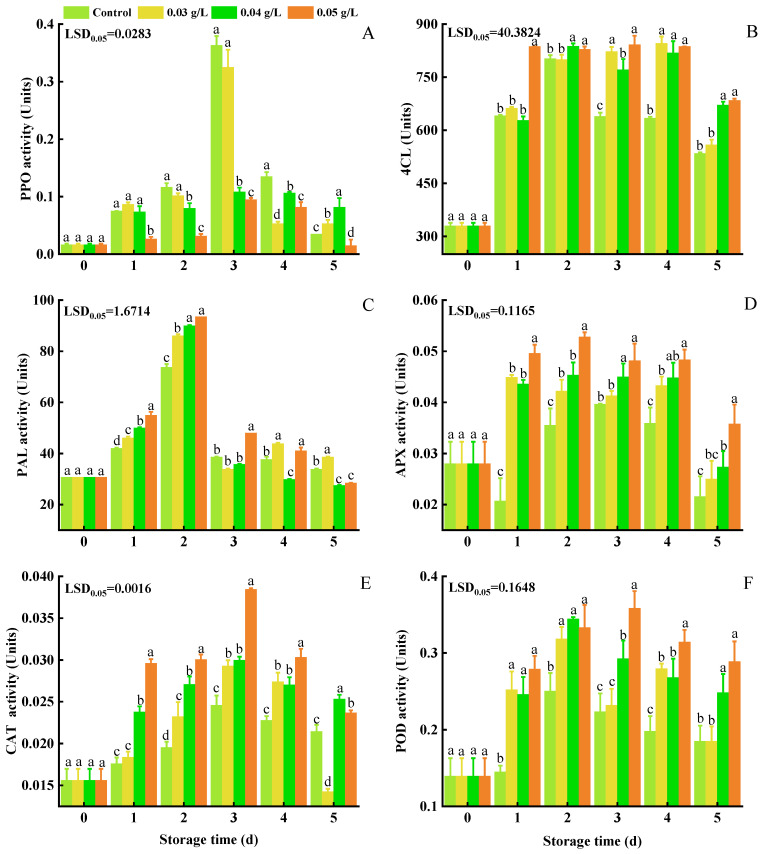
Effect of PA on the activity of PPO (**A**), 4CL (**B**), PAL (**C**), APX (**D**), CAT (**E**), and POD (**F**) in mini-Chinese cabbage during five days of storage at 25 °C. Data represent the mean ± SD (*n* = 9). Different letters above the bars within each time point indicate a significant difference between the designated treatment groups at *p* < 0.05 as determined by an LSD test.

**Figure 4 foods-12-01105-f004:**
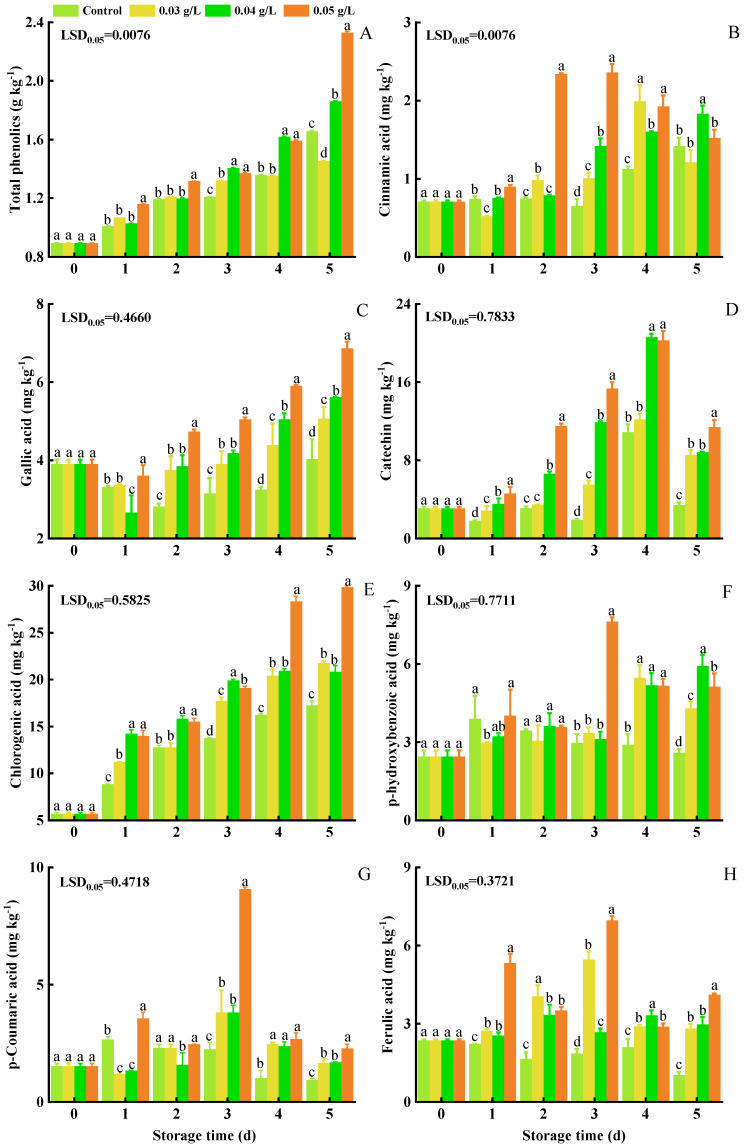
Effect of PA on the content of total phenolics (**A**), cinnamic acid (**B**), gallic acid (**C**), catechin (**D**), chlorogenic acid (**E**), p-Hydroxybenzoic acid (**F**), p-Coumaric acid (**G**), and ferulic acid (**H**) in mini-Chinese cabbage during five days of storage at 25 °C. Data represent the mean ± SD (*n* = 9). Different letters above the bars within each time point indicate a significant difference between the designated treatment groups at *p* < 0.05 as determined by an LSD test.

**Figure 5 foods-12-01105-f005:**
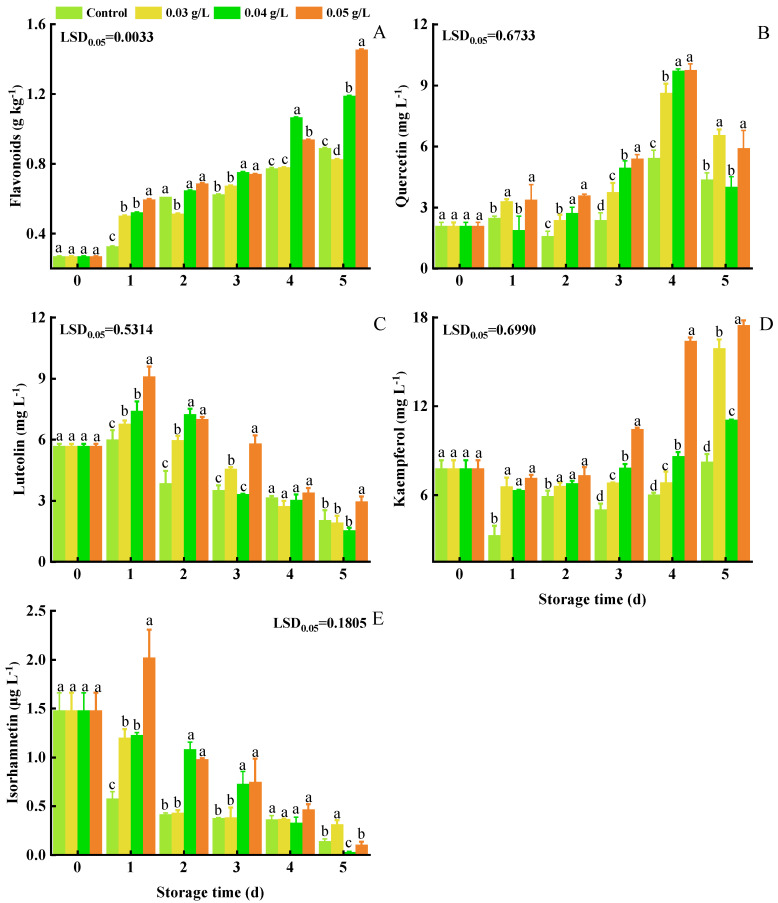
Effect of PA on the content of flavonoids (**A**), quercetin (**B**), luteolin (**C**), kaempferol (**D**), and isorhamnetin (**E**) in mini-Chinese cabbage during five days of storage at 25 °C. Data represent the mean ± SD (*n* = 9). Different letters above the bars within each time point indicate a significant difference between the designated treatment groups at *p* < 0.05 as determined by an LSD test.

**Figure 6 foods-12-01105-f006:**
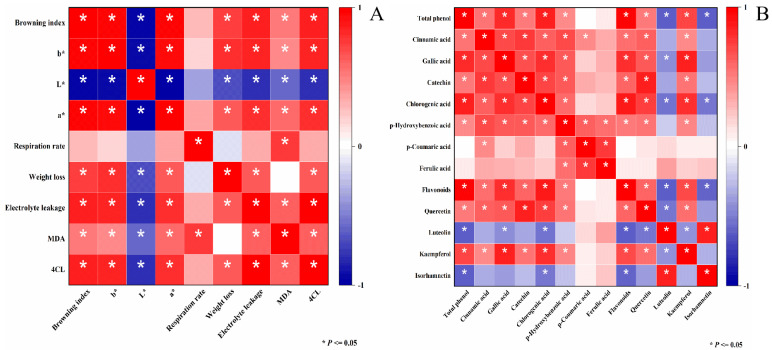
Correlation between different parameters measured in mini-Chinese cabbage stored for five days at 25 °C as determined by Pearson’s coefficients (**A,B**). * indicates a significant correlation at *p* < 0.05. The color indicates the level of the positive (red) or negative (blue) correlation as indicated in the color key at the side of each figure.

## Data Availability

Data available upon request to corresponding author.

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
