# Peer review of "Palmitic Acid Regulation of Stem Browning in Freshly Harvested Mini-Chinese Cabbage (Brassica pekinensis (Lour.) Rupr.)"

_foods, 2023, doi:10.3390/foods12051105_

Round 1

Reviewer 1 Report

Specific comments

            This paper analyzed palmitic acid's effect on stem browning in freshly harvested Mini-Chinese cabbage. The authors understandably wrote the manuscript. However, the manuscript had some minor errors, which are listed below. I also have some more doubts about this work. Why did the authors not analyze the biochemical traits, browning index, and other traits under >0.05gm of PA concentration? What basis did the authors select the three PA concentrations for this experiment? Why did the authors not analyze the PA, pigment, and protein contents? If authors have the sample, authors should include these analysis results. Authors should include all information in the manuscript. The discussion part needs to expand based on the previous results. Also, the authors should write the results between the days. Furthermore, authors can increase the contents of the correlation analysis part based on their results.

Minor comments

1.      L. No. 15. "Palmitic acid" should be changed into "PA"

2.      Write Mini-Chinese Cabbage instead of freshly harvested Mini-Chinese Cabbage in the keywords.

3.      Authors should write the full form of PPO in L. No. 42

4.      L. No. 55; authors mentioned that the use of PA to regulate browning had been investigated. Kindly complete this sentence. Had been investigated in which plants or animals?

5.      L. No. 73, min o mini?

6.      No need to include the botanical name of cabbages in L. No. 73.

7.      Is “Xiaogiao” cultivar name?

8.      Figure 1a is not included

9.      The authors mentioned a short form of a figure ("Fig") inside the manuscript. But authors wrote the full form of the figure in all the figure legends.  

10.  Did authors receive negative results for color? The authors should check the Figure 1D results.

11.  The authors should start all the graphs (Y-axis range) from 0 onwards because I could not understand the results.  

Author Response

Reviewer 1:

   This paper analyzed palmitic acid's effect on stem browning in freshly harvested Mini-Chinese cabbage. The authors understandably wrote the manuscript. However, the manuscript had some minor errors, which are listed below. I also have some more doubts about this work. 1. Why did the authors not analyze the biochemical traits, browning index, and other traits under >0.05gm of PA concentration? 2. What basis did the authors select the three PA concentrations for this experiment? 3. Why did the authors not analyze the PA, pigment, and protein contents? If authors have the sample, authors should include these analysis results. Authors should include all information in the manuscript. The discussion part needs to expand based on the previous results. Also, the authors should write the results between the days. Furthermore, authors can increase the contents of the correlation analysis part based on their results.

Answer: Thank you very much for your comments. 1. We made a pre-test and the result showed that 0.05 g L-1 was more better than other > 0.05 g L-1 groups. 2. We made a pre-test and we found these concentrations were better than other group. 3. Thank you for the above suggestion. The result was focused on the browning, thus, we measured some substances about browning. PA, pigment, and protein contents were not the most important points in our study.

Minor comments:

  1. No. 15. "Palmitic acid" should be changed into "PA".

Answer: Thank you very much for your comments. The “Palmitic acid” was changed into “PA” at lines 15.

  1. Write Mini-Chinese Cabbage instead of freshly harvested Mini-Chinese Cabbage in the keywords.

Answer: Thank you very much for your comments. We have changed “freshly harvested Mini-Chinese Cabbage” as “Mini-Chinese Cabbage” in the keywords.

  1. Authors should write the full form of PPO in L. No. 42.

Answer: Thank you very much for your comments. We added the full form of PPO at lines 42-43.

  1. No. 55; authors mentioned that the use of PA to regulate browning had been investigated. Kindly complete this sentence. Had been investigated in which plants or animals?

Answer: Thank you very much for your comments. We have completed this sentence at lines 55-56.

  1. L. No. 73, min o mini?

Answer: Thank you very much for your comments. We have changed “min” as “mini” at line 74.

  1. No need to include the botanical name of cabbages in L. No. 73.

Answer: Thank you very much for your comments. We deleted the botanical name of cabbages.

  1. Is “Xiaogiao” cultivar name?

Answer: Thank you very much for your comments. “Xiaogiao” is the cultivar name.

  1. Figure 1a is not included.

Answer: Thank you very much for your comments. We have added the Figure 1A at line 212.

  1. The authors mentioned a short form of a figure ("Fig") inside the manuscript. But authors wrote the full form of the figure in all the figure legends.  

Answer: Thank you very much for your comments. We modified all of the “Fig” as figure in the manuscript.

  1. Did authors receive negative results for color? The authors should check the Figure 1D results.

Answer: Thank you very much for your comments. Figure 1D is the value of a*. As we know, a* represents the red and green values. Positive values indicate red and negative values indicate green. When the stem of cabbage showed a browning phenomenon the color is more close to red rather than green. Thus, control group showed a higher positive value in our data.

  1. The authors should start all the graphs (Y-axis range) from 0 onwards because I could not understand the results.  

Answer: Thank you very much for your comments. We according the method of Aleman et al. (2023) and Díez-Betriu et al. (2023) to draw the figures. Their figures (Y-axis range) did not start from 0 onwards. And some number was too large we can’t start from 0 onwards in our data.

  1. Aleman, R.S.; Marcía, J.A.; Montero-Fernández, I.; King, J.; Pournaki, S.K.; Hoskin, R.T.; Moncada, M. Novel Liquor-Based Hot Sauce: Physicochemical Attributes, Volatile Compounds, Sensory Evaluation, Consumer Perception, Emotions, and Purchase Intent. Foods 2023, 12, 369. https://doi.org/10.3390/foods12020369.
  2. Díez-Betriu, A.; Bustamante, J.; Romero, A.; Ninot, A.; Tres, A.; Vichi, S.; Guardiola, F. Effect of the Storage Conditions and Freezing Speed on the Color and Chlorophyll Profile of Premium Extra Virgin Olive Oils. Foods 2023, 12, 222. https://doi.org/10.3390/foods12010222.

Reviewer 2 Report

The authors of the manuscript titled " Palmitic acid regulation of stem browning in freshly harvested 1 mini-Chinese cabbage (Brassica pekinensis (Lour.) Rupr.)" have made a great effort. Still, there is a significant comment in the manuscript. So please check the attached file and answer all the questions related to the words highlighted. 

Round 2

Reviewer 1 Report

The authors addressed all the comments appropriately. So, accept the manuscript in the present form 

Author Response

Thank you very much for your comments.

Reviewer 2 Report

The authors of the manuscript titled " Palmitic acid regulation of stem browning in freshly harvested 1 mini-Chinese cabbage (Brassica pekinensis (Lour.) Rupr.)" have made a great effort and answered most of the comments and I recommend accepting the manuscript. Still, there are a few comments on the following:- 

1- in line 81 -- what is the concentration of methanol used? this question was not answered, please confirm. 

2- in my comment on the manufacturer of the chemicals and instruments please put them in the form of ( manufacturer, town, and country) for example ((Aladdin, AR, Town, and Country ) Also do the same for all chemicals and instruments.

3- in line 150---- what is the VWD ?? 

4- I suggest putting the chromatogram and the polyphenolic calibration standard curve in the supplementary materials and referring to them in the results section.   

Author Response

Point-by-Point Responses to comments made by the reviewers and the editor

Manuscript ID: foods-2213180

Palmitic acid regulation of stem browning in freshly harvested mini-Chinese cabbage (Brassica pekinensis (Lour.) Rupr.)

Reviewer 2:

The authors of the manuscript titled " Palmitic acid regulation of stem browning in freshly harvested 1 mini-Chinese cabbage (Brassica pekinensis (Lour.) Rupr.)" have made a great effort and answered most of the comments and I recommend accepting the manuscript. Still, there are a few comments on the following:-

1- in line 81 -- what is the concentration of methanol used? this question was not answered, please confirm.

Answer: Thank you very much for your comments. The concentration of methanol is 100 % at lines 156.

2- in my comment on the manufacturer of the chemicals and instruments please put them in the form of ( manufacturer, town, and country) for example ((Aladdin, AR, Town, and Country ) Also do the same for all chemicals and instruments.

Answer: Thank you very much for your comments. We modified the manufacturer of the chemicals and instruments as the form of ( manufacturer, town, and country) at the part of materials and methods.

3- in line 150---- what is the VWD ??

Answer: Thank you very much for your comments. We changed the “VWD” as “UV- detector” at lines 155.

4- I suggest putting the chromatogram and the polyphenolic calibration standard curve in the supplementary materials and referring to them in the results section.

Answer: Thank you very much for your comments. We added a supplementary file 1.